# Cross-sectional seroprevalence surveys of SARS-CoV-2 antibodies in children in Germany, June 2020 to May 2021

Anna-Lisa Sorg [1,2], Leon Bergfeld[3], Marietta Jank[4], Victor Corman [3], Ilia Semmler[3], Anna Goertz[4], Andreas Beyerlein [1], Eva Verjans[5], Norbert Wagner[5], Horst Von Bernuth [6], Fabian Lander[7], Katharina Weil[8], Markus Hufnagel[9], Ute Spiekerkoetter [9], Cho-Ming Chao[10,11], Lutz Naehrlich[11], Ania Carolina Muntau[12], Ulf Schulze-Sturm [12], Gesine Hansen[13], Martin Wetzke[13], Anna-Maria Jung[14], Tim Niehues[15], Susanne Fricke-Otto[15], Ulrich Von Both [ ], Johannes Huebner [16], Uta Behrends[17], Johannes G. Liese[18], Christian Schwerk [4], Christian Drosten [3,19], Ruediger Von Kries [1,19 ✉] & Horst Schroten[4,19]

The rate of SARS-CoV-2 infections in children remains unclear due to many asymptomatic cases. We present a study of cross-sectional seroprevalence surveys of anti-SARS-CoV-2 IgG in 10,358 children recruited in paediatric hospitals across Germany from June 2020 to May 2021. Seropositivity increased from 2.0% (95% CI 1.6, 2.5) to 10.8% (95% CI 8.7, 12.9) in March 2021 with little change up to May 2021. Rates increased by migrant background (2.8%, 4.4% and 7.8% for no, one and two parents born outside Germany). Children under three were initially 3.6 (95% CI 2.3, 5.7) times more likely to be seropositive with levels equalising later. The ratio of seropositive cases per recalled infection decreased from 8.6 to 2.8. Since seropositivity exceeds the rate of recalled infections considerably, serologic testing may provide a more valid estimate of infections, which is required to assess both the spread and the risk for severe outcomes of SARS-CoV-2 infections.

[1] Institute of Social Paediatrics and Adolescent Medicine, Division of Paediatric Epidemiology, Ludwig-Maximilians-University Munich, 80336 Munich, Germany. [2] University Children's Hospital, Eberhard Karls University, 72076 Tuebingen, Germany. [3] Institute of Virology, Charité Universitätsmedizin Berlin, 10117 Berlin, Germany. [4] Paediatric Infectious Diseases, Department of Paediatrics, Medical Faculty Mannheim, Heidelberg University, 68167 Mannheim, Germany. [5] Department of Paediatrics, Medical Faculty, University Hospital RWTH Aachen, 52074 Aachen, Germany. [6] Department of Paediatric Respiratory Medicine, Immunology, and Critical Care Medicine, Charité Universitätsmedizin Berlin, 13353 Berlin, Germany. [7] Department of Paediatrics, University Hospital, and Medical Faculty Carl Gustav Carus, Technical University (TU) Dresden, 01307 Dresden, Germany. [8] Department of General Paediatrics, Neonatology, and Paediatric Cardiology, Medical Faculty, University Hospital, Heinrich-Heine-University Düsseldorf, 40225 Düsseldorf, Germany. [9] Department of Paediatrics and Adolescent Medicine, University Medical Centre, Medical Faculty, University of Freiburg, 79106 Freiburg, Germany. [10] University Medical Centre Rostock, Department of Paediatrics, University of Rostock, 18057 Rostock, Germany. [11] Universities of Giessen and Marburg Lung Centre, German Centre of Lung Research, Department of Paediatrics, Justus-Liebig-University Giessen, 35392 Giessen, Germany. [12] University Children's Hospital, University Medical Centre Hamburg- Eppendorf, 20246 Hamburg, Germany. [13] Centre for Paediatrics and Adolescent Medicine, Hannover Medical School, Excellence Cluster RESIST, Deutsche Forschungsgemeinschaft (DFG), EXS 2155, 30625 Hannover, Germany. [14] Department of General Paediatrics, Neonatology, Children's Hospital Medical Centre, Saarland University Hospital, 66421 Homburg/Saar, Germany. [15] Department of Paediatrics, Helios Klinikum Krefeld, 47805 Krefeld, Germany. [16] Dr von Hauner Children's Hospital, University Hospital, Ludwig-Maximilians-University, 80337 Munich, Germany. [17] Department of Paediatrics, Faculty of Medicine, Technical University Munich, 80804 Munich, Germany. [18] University Hospital of Wuerzburg, Department of Paediatrics, Division of Paediatric Infectious Diseases, 97080 Wuerzburg, Germany. [19] These authors contributed equally: Christian Drosten, Ruediger von Kries, Horst Schroten. ✉email: Ruediger.kries@med.uni-muenchen.de

With advanced COVID-19 vaccination of the elderly and adults, the level of infection-derived immunity and group susceptibility of children becomes an urgent issue. The contribution of children to the transmission of Severe Acute Respiratory Syndrome Coronavirus 2 (SARS-CoV-2) at the population level is influenced by vaccination rates in other age groups and may change with the emergence of viral variants with higher viral loads and longer viral shedding[1]. A German national registry reported 1647 hospitalised children during January 2020 to May 2021 with laboratory-confirmed SARS-CoV-2 infections, of whom about 20% required SARS-CoV-2 associated therapy[2]. To date, available evidence indicates that children develop the less severe diseases than adults[3]. Because the presentation of COVID-19 in children is mainly asymptomatic or mild, the proportion of underreported cases in this age group is likely to be particularly high. Unreported cases, however, may contribute as transmitters to community outbreaks[4]. Additionally, knowledge of the proportion and age range of seropositive children is essential for the design of control and vaccination strategies.

For more than 1 year from the pandemic onset, Germany followed relatively rigid strategies to control incidence, involving full or partial closures of educational and childcare facilities. Seroprevalence of children in Germany may thus provide an important reference for comparison with countries that followed different approaches, particularly in school and education settings. Previous seroprevalence studies in Germany indicated that reported case numbers underestimated the rates of infection in children[5–7]. Though previous studies for example from the US, UK or Italy observed increased infection rates in groups with migrant background[8–10], no data from Germany are available. It is interesting to assess migrant background since it might be a surrogate for effects of various determinants such as higher risk occupations, and differences in mobility patterns or household sizes.

Previous studies were temporally and regionally limited, as is often the case for surveys on the seroprevalence of SARS-CoV-2, especially in children[11–13]. Therefore, to date there are no comprehensive data available enabling to study temporal trends and potentially related factors of COVID-19 seroprevalence of children in Germany.

In this study, we assessed the temporal course of seroprevalence of SARS-CoV-2 antibodies of children in Germany employing nationwide, multicentre, cross-sectional seroprevalence surveys. We report results of a 1-year observation period from June 2020 to May 2021, prior to the onset of general recommendations for COVID-19 vaccination for children. Differences in age groups, migrant background and prior recognition of SARS-CoV-2 infections were analysed.

## Results

**Characteristics of the study population.** Information on 10,358 patients, recruited between 1 June 2020, and 31 May 2021, was available for analysis with a monthly recruiting average of 863 (±SD 220) participants across all 14 study centres. Fig. 1 presents the proportional contribution to the total study recruitment by location.

Patients' characteristics potentially associated with seroprevalence are listed in Table 1. The median age was 10.3 years (Interquartile Range (IQR) 5.3, 14.3 years). Children under three accounted for 14.3% ($n = 1437$) of the study sample, 45.6% ($n = 4727$) of the participants were three to 12 years old, and 40.1% ($n = 4152$) 12–17-year old. Overall, 37.6% had a migrant background (one [11.7%] or both parents [26.0%] with country of origin outside Germany) with some variability over the recruitment months (range 32.6–41.3%). In this hospital-based study, nearly 60% of respondents reported pre-existing conditions.

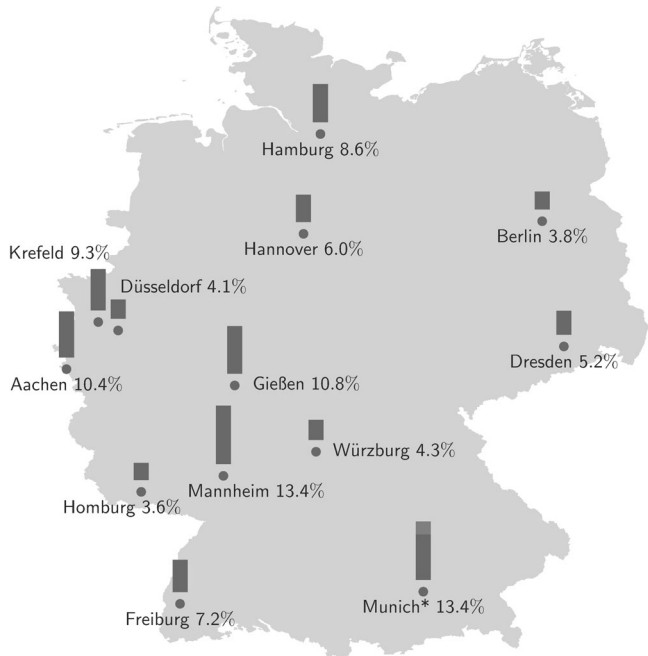

**Fig. 1 Distribution of study centres.** 14 children's hospitals, spread all over Germany, took part in the SARS-CoV-2 study. In total 10,358 pedaitric patients participated. The proportion of recruitment per study centre range from 3.6% (373/10358) to 13.4% (1387/10358). *in Munich, two separate study centres recruited—Paediatric Department of the Ludwig-Maximilians-University 10.3% and the Technical University Munich 3.1%.

**Seroprevalence of SARS-CoV-2 antibodies.** Overall, SARS-CoV-2 antibodies with OD ratio ≥ 1.1 were detectable in 461 of the 10,358 (4.5%) children. Besides determinants expected to be significantly associated with increased seropositivity per se such as a 'test of SARS-CoV-2 infection in the 'past' or a 'history of respiratory 'diseases', age group, country of origin of the parents and language spoken in the family were found to be significantly associated with seropositivity, while sex and pre-existing medical conditions were not (Table 1).

Of seropositive children with information of previous respiratory infections, 22.6% ($n = 96/424$) had one or more respiratory tract infections with symptoms such as fever or shortness of breath since March 2020, as opposed to 13.9% in children without SARS-CoV-2 antibodies ($n = 1336/9563$, $p = <0.0001$).

In Germany, the number of infections and measures to contain the pandemic varied depending on the federal state. Supplementary Fig. 1 shows the point estimates and 95% confidence of the prevalence of SARS-CoV-2 IgG in the different stages of the COVID-19 pandemic in Germany stratified by study centre.

**Time course of SARS-CoV-2 seroprevalence.** The study started at the end of the first pandemic wave in Germany with an average seroprevalence of SARS-CoV-2 IgG antibodies of 2.0% (95% CI 1.6, 2.5) in June to September 2020. From October 2020 onwards, there was an increase to 10.8% (95% CI 8.7, 12.9) until March 2021 with no further major increase to the end of observation in May 2021 (Fig. 2).

The prevalence of SARS-CoV-2 antibodies was significantly higher in children younger than 3 years (5.4%, $n = 28/519$) than in older children (3–11 years, 1.4%, $n = 28/1955$; >12 years, 1.7%, $n = 30/1770$; $p < 0.0001$) during June to September 2020, resulting in an odds ratio (OR) of 3.61 (95% CI 2.27; 5.72) to test seropositive for the comparison of <3-year old with older children during this time (Fig. 3). The strength of this association was unchanged if

| Table 1 Characteristics of the study population potentially influencing seroprevalence. | | | | |
|---|---|---|---|---|
| | $N^*$ | %[a] | Proportion of children with SARS-CoV-2 IgG antibodies | p-value[b] |
| Sex | 10,338 | | | 0.48 |
| Male | | 5110 | 49.4 | 4.3 |
| Female | | 5228 | 50.6 | 4.6 |
| Age group | 10,358 | | | <0.0001* |
| < 3 years of age | | 1479 | 14.3 | 6.6 |
| 3–11 years of age | | 4727 | 45.6 | 3.9 |
| 12–17 years of age | | 4152 | 40.1 | 4.4 |
| Country of origin of parents | 9922 | | | <0.0001* |
| Germany (both parents) | | 6187 | 62.4 | 2.8 |
| Germany (one parent) | | 1157 | 11.7 | 4.4 |
| Outside Germany (both parents) | | 2578 | 26·0 | 7.8 |
| Language spoken in the family | 9871 | | | <0.0001* |
| German | | 8913 | 90.3 | 3.9 |
| Other Language | | 958 | 9.7 | 8.8 |
| Reason for hospitalisation | 7623 | | | 0.02 |
| Elective treatment | | 1722 | 22.6 | 5.3 |
| Routine check-up | | 2953 | 38.7 | 4.0 |
| Referral for inpatient evaluation or parent/patient education | | 825 | 10.8 | 3.8 |
| Emergency | | 1449 | 19.0 | 6.0 |
| Other | | 674 | 8.8 | 4.8 |
| Respiratory infection or pneumonia as reason for hospitalisation | 9867 | | | 0.19 |
| Yes | | 291 | 2.9 | 5.8 |
| No | | 9576 | 97.1 | 4.3 |
| Test of SARS-CoV-2 infection in the past[c] | 10,123 | | | <0.0001* |
| Yes | | 4073 | 40.2 | 7.2 |
| No | | 6050 | 59.8 | 2.5 |
| SARS-CoV-2 test result in the past[c] | 4073 | | | <0.0001* |
| Positive | | 167 | 4.1 | 71.3 |
| Negative | | 3906 | 95.9 | 4.4 |
| History of respiratory diseases since March 2020 | 9987 | | | <0.0001* |
| Yes | | 1432 | 14.3 | 6.7 |
| No | | 8555 | 85.7 | 3.8 |
| History of pneumonia since March 2020 | 10,056 | | | <0.0001* |
| Yes | | 229 | 2.3 | 10.5 |
| No | | 9827 | 97.7 | 4.2 |
| History of hospitalisation due to pneumonia since March 2020 | 9268 | | | 0.0001* |
| Yes | | 140 | 1.5 | 10.7 |
| No | | 9128 | 98.5 | 4.2 |
| Past medical history (pre-existing conditions) | 10,076 | | | 0.98 |
| Yes | | 5788 | 57.4 | 4.4 |
| No | | 4288 | 42.6 | 4.4 |
| Selected pre-existing conditions | | | | |
| Asthma | 8941 | | | 0.96 |
| Yes | | 627 | 7.0 | 4.5 |
| No | | 9314 | 93.0 | 4.5 |
| Mucoviscidosis | 8895 | | | 0.16 |
| Yes | | 172 | 1.9 | 2.3 |
| No | | 8723 | 98.1 | 4.6 |
| Bronchopulmonary dysplasia (BPD) | 8810 | | | 0.07 |
| Yes | | 81 | 0.9 | 8.6 |
| No | | 8729 | 99.1 | 4.5 |
| Heart disease/heart defect | 8961 | | | 0.10 |
| Yes | | 430 | 4.8 | 6.1 |
| No | | 8531 | 95.2 | 4.4 |
| Haematological/oncological disease | 8883 | | | 0.38 |
| Yes | | 489 | 5.5 | 3.7 |
| No | | 8394 | 94.5 | 4.5 |
| Neurological/neuromuscular disease | 8873 | | | 0.008 |
| Yes | | 712 | 8.0 | 6.5 |
| No | | 8161 | 92.0 | 4.3 |
| Congenital or acquired immunodeficiency | 8767 | | | 0.42 |
| Yes | | 211 | 2.4 | 3.3 |
| No | | 8556 | 97.6 | 4.5 |

**Table 1 (continued)**

| | N* | %ᵃ | Proportion of children with SARS-CoV-2 IgG antibodies | | p-valueᵇ |
|---|---|---|---|---|---|
| *Autoimmune disease* | 8879 | | | | 0.01 |
| Yes | | 962 | 10.8 | 2.9 | |
| No | | 7917 | 89.2 | 4.7 | |
| *Metabolic disease* | 6759 | | | | 0.57 |
| Yes | | 528 | 7.8 | 5.5 | |
| No | | 6231 | 92.2 | 4.9 | |
| *Gastrointestinal disease* | 6721 | | | | 0.30 |
| Yes | | 615 | 9.2 | 4.1 | |
| No | | 6106 | 90.9 | 5.0 | |
| *Chronic renal disease* | 6701 | | | | 0.60 |
| Yes | | 265 | 4.0 | 5.7 | |
| No | | 6436 | 96.0 | 4.9 | |

*Difference in the absolute number of recruits of 10358 is due to unanswered questions.
ᵃPercentage not adding to 100% is explained by rounding.
ᵇTwo-sided exact p-values for Pearson chi-square (p-values marked with * are <0.05 after Bonferroni correction) for the association to SARS-CoV-2 IgG antibodies.
ᶜQuestionnaire did not specifiy the applied test method.
N number of participants with available information, SARS-CoV-2 severe acute respiratory syndrome coronavirus 2, IgG immunoglobulin G.

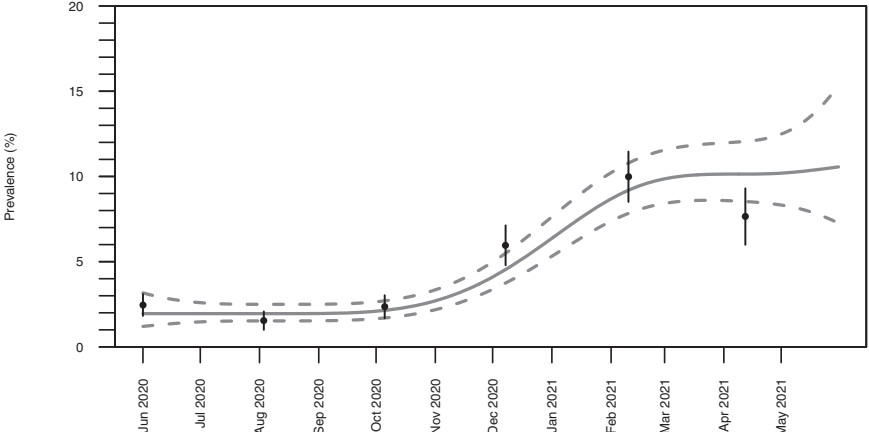

**Fig. 2 Trends in SARS-CoV-2 seroprevalence from June 2020 to May 2021 in children in Germany.** Two-month-average point estimates of the prevalence of Severe Acute Respiratory Syndrome Coronavirus 2 immunoglobulin G antibodies as determined by an enzyme-linked immunosorbent assay in blood samples from 10,358 paediatric study participants. The black dots display the respective point estimates of the prevalences and the wkiskers (lines at the black dots) the 95% confidence intervals of the point estimates. The predicted probability according to a b-spline regression model (grey solid line) with 95% confidence band (grey dashed lines).

children ≤ 6 months were excluded to rule out the possible influence of maternal antibodies (OR 3.56 (95% CI 2.19, 5.79)). This association decreased in the subsequent pandemic phase (October 2020 to February 2021: OR 1.43 (95% CI 1.02, 2.02)) since the seroprevalence increased at a higher rate in children over three compared to those under three (Fig. 3). In March to May 2021, this age group difference was no longer apparent (OR 1.00 (95% CI 0.64, 1.56)).

The multivariable logistic regression model confirmed an increased seropositivity rate for migrant background with an OR of 1.61 (95%CI 1.16; 2.22) for children with one parent from abroad and an OR of 2.90 (95% CI 2.35; 3.59) for children with both parents from abroad compared to children with both parents having Germany as country of origin. Compared to 12–17-year-old children, age under three was associated with a significantly higher risk of seropositivity with an OR of 1.39 (95% CI 1.06; 1.82), while the group of 3–11-year old children had a slightly lower risk with an OR of 0.82 (95% CI 0.66; 1.02). Accordingly, the temporal courses differed when stratified by migrant background (Supplementary Fig. 2) and age group (Supplementary Fig. 3). The overall curve of the temporal course

adjusted for migrant background and age group was almost identical to the unadjusted curve, visualised in a partial residual plot (Supplementary Fig. 4). Interaction terms of country of origin with one parent from abroad and country of origin of both parents from abroad with time were statistically not significant (p = 0.19, respectively, p = 0.09). In contrast, there was a significant interaction for the age groups and time (<3-year old: p = 0.0003, respectively, 3–11-year old: p = 0.04 with 12–17-year old as reference).

**Associations with previous SARS-CoV-2 testing.** Participants with recalled previous SARS-CoV-2 testing reported significantly more often a history of respiratory disease, pneumonia or hospitalisation due to pneumonia, and the children were more likely to have a pre-existing medical history (Supplementary Table 1). A previous test was more often reported in children with heart diseases, haematological/oncological disease, neurological/neuromuscular diseases, gastrointestinal or chronic renal diseases. In contrast, previous testing was not associated with children's sex, age or migrant background.

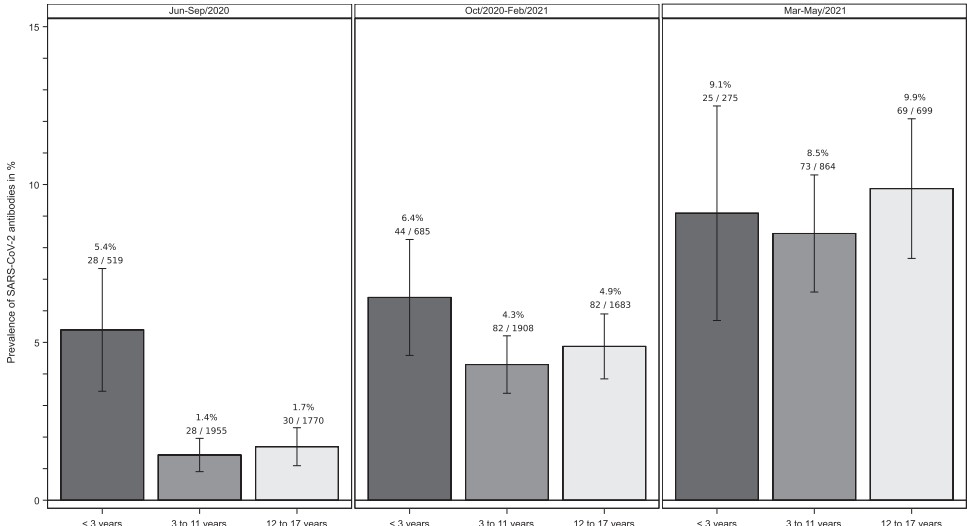

**Fig. 3 Age group-specific seroprevalence etimates in different phases of the COVID-19 pandemic in Germany.** Prevalence of Severe Acute Respiratory Syndrome Coronavirus 2 (SARS-CoV-2) immunoglobulin G antibodies as determined by an enzyme-linked immunosorbent assay in blood samples from in total 10,358 paediatric study participants in the different stages of the COVID-19 pandemic in Germany stratified by age category. The height of the boxes displays the respective point estimates of the prevalences and the wiskers indicates the upper and lower 95% confidence limits of these point estimates.

**Table 2 Ratio of children with previously reported positive SARS-CoV-2 test‡ to all children with SARS-CoV-2 IgG antibodies, as determined by an Enzyme-Linked Immunosorbent Assay (ELISA), in different phases of the pandemic according to the date of study recruitment.**

| | Previously reported positive SARS-CoV-2 test[a] (known infection) | Total-SARS-CoV-2 positive ELISA (IgG antibodies) | Known: total |
|---|---|---|---|
| *All age groups* | | | |
| *Total observation period* | 119 | 461 | 1: 3.9 |
| Jun–Sep 2020 | 10 | 86 | 1: 8.6 |
| Oct 2020–Feb 2021 | 49 | 208 | 1: 4.2 |
| Mar–May 2021 | 60 | 167 | 1: 2.8 |
| A | < 3 years | | |
| *Total observation period* | 12 | 97 | 1: 8.1 |
| Jun–Sep 2020 | 1 | 28 | 1: 28 |
| Oct 2020–Feb 2021 | 7 | 44 | 1: 6.3 |
| Mar–May 2021 | 4 | 25 | 1: 6.3 |
| B | 3–11 years | | |
| *Total observation period* | 41 | 183 | 1: 4.5 |
| Jun–Sep 2020 | 3 | 28 | 1: 9.3 |
| Oct 2020 – Feb 2021 | 14 | 82 | 1: 5.9 |
| Mar–May 2021 | 24 | 73 | 1: 3.0 |
| C | 12–17 years | | |
| *Total observation period* | 66 | 181 | 1: 2.7 |
| Jun–Sep 2020 | 6 | 30 | 1: 5 |
| Oct 2020–Feb 2021 | 28 | 82 | 1: 2.9 |
| Mar–May 2021 | 32 | 69 | 1: 2.2 |

[a]Questionnaire did not specify the applied test method.
*SARS-CoV-2* severe acute respiratory syndrome coronavirus 2.

Overall, there were 4073 participants with information on previous SARS-CoV-2 test results, of whom 167 (4.1%) had received a positive SARS-CoV-2 test result. In 25.8% ($n = 119/461$) of the seropositive participants an infection with SARS-CoV-2 had been previously diagnosed, which corresponds to 3.9 cases per recalled infection over the entire observation period (Table 2), while there were reports of previous SARS-CoV-2 infections in only 0.5% ($n = 48/9897$) of the seronegative participants.

The number of seropositive cases per recalled infection decreased from 8.6 in June to September 2020 to 2.8 in March to May 2021 (Table 2). A similar decrease was observed across all

age groups. In each part of the observation period, the detection rates were lower in the younger age groups, with rates of 1: 6.3 for children <3 years compared to 1: 3.0 for children aged 3–11 years and 1: 2.2 for children aged 12–17 years from March to May 2021, respectively (Table 2 A, C).

**Prevalence of neutralising antibodies**. 143 of the 252 sera, additionally tested by PRNT, showed an ELISA OD ratio ≥1.1 and 109 an OD ratio <1.1. Neutralising antibodies were found in 55/252 (21.8%) sera. 94.5% of PRNT-50 positive sera showed an OD ratio ≥1.1 and 0.05% of PRNT-50 positive were within the ELISA OD ratio borderline range (0.8–1.1), none of the sera with

OD ratio below 0.8 tested positive for neutralising antibodies (Supplementary Fig. 5A, B).

**ELISA threshold optimisation.** ROC analysis yielded different optimal cut-off values for the ELISA (see Supplementary Methods), accounting for different absolute estimates of seroprevalence. The temporal trend of seroprevalence according to b-spline regression models was similar for all three tested thresholds (Supplementary Fig. 6). The manufacturer-recommended threshold at OD ratio 1.1 may thus be a valid and useful classifier in paediatric serosurveys, additionally allowing comparison with adult serosurveys.

**External validity of the results.** Age and sex distribution in our study sample compared to the general German population of children ≤17 years in 2020 was slightly shifted towards older ages, more pronounced in the female group (Supplementary Fig. 7).

Two-month seroprevalence estimates, standardised for migrant background, age groups, and study sites, were similar compared to crude seroprevalence estimates with overlapping confidence intervals (Supplementary Table 2). External validity is supported by these comparable estimates.

## Discussion

This study reveals a seroprevalence of 10.8% in children by March 2021, admitted to German paediatric hospitals for various reasons, with no major change up to May 2021. The steepest increase was observed in the second wave of the pandemic. The time trend in seropositivity rates varied in different age groups and by migrant background. Whereas seroprevalence studies are thought to reflect the true infection activity at the population level, as opposed to measurements of point prevalence by RT-PCR, some caution is required when comparing the present results against whole population assessments.

A recent seroprevalence study in Bavaria, a federal state of Germany, found seroprevalence estimates in 1–5 and 6–10-year-old children of 5.6% and 8.4% in February 2021, respectively[7]. When we applied these age groups to our data, we found corresponding estimates of 9.8% and 7.8%. Therefore, while the prevalence estimates for 6–10-year-old children agreed well between the two studies, there seems to be a higher seroprevalence in young children in the present study. Differences in the utilisation of medical services (hospital versus private offices) could contribute to this discrepancy.

One explanation for increased seroprevalence in younger children from June to September 2020 as observed in our data may be a different role of household transmission. We were able to rule out an alternative explanation of infants carrying specific antibodies from their mothers, as excluding infants younger than 6 months did not change the results. Young children are likely to have closer contact with adult virus carriers than older children in the family. In a meta-analysis, a similar phenomenon was seen for married couples (higher attack rate)[14]. As the first wave was characterised by relatively strict closures of childcare and educational settings in Germany, changes in seroprevalence may reflect a gradual change of epidemic patterns, with predominantly household-based acquisition during the first wave and institutional- or community-based acquisition in later periods. There are several other studies, however mainly regional and often with small sample sizes, in which similar effects can be observed. A study in Seattle, US, with a design similar to ours, observed a low seroprevalence in children in general, but an increased prevalence in 0–4-year-old children[12]. Another US hospital-based study, in Arkansas, demonstrated a strong predominance of seroprevalence in 1–4-year-old children from April to October 2020, a time when schools were closed, followed by a re-distribution toward older

groups in late 2020[15]. In Croatia, children under 10 had higher seroprevalence than other age groups after the first wave, with the pattern reversed after the second wave[16]. In Madrid after the first wave, children aged 0–4 years had a higher seroprevalence compared to the other age groups[17]. A comprehensive serosurvey in Wuhan, China, conducted after the initial outbreak with subsequent community lockdown in early 2020, identified an increased seroprevalence in children aged 0–5 years as compared to children of all other age groups[18]. Some caution may thus be necessary when interpreting population-based studies. The resolution of age categories, and the timing of the first wave, may be critical to analyse patterns of infection across age tiers. Contact restrictions in place during study periods may have contributed to the broad impression of significantly lower infection levels in children, while settings with less strict containment measures showed similar seroprevalence in children and adults. For instance, age groups <5, 5–18 and 19–45 years all had similar seroprevalence in South Africa by late 2020[19]. Based on analyses of attack rates and contact patterns, a study based on contact tracing data during the 2020 outbreak in China concluded that children may be slightly less susceptible than adults, but this difference may be entirely compensated by more intense contact behaviour[20].

Clear population differences in SARS-CoV-2 infection have been reported since the pandemic outbreak[21]. The differences occurred in terms of occupational or age groups, and when stratified by ethnicity[22]. Positivity rates in non-Hispanic, Black, and Hispanic people aged <18 years were 2.4 and 4.3 times increased, respectively, compared to the rate within the white ethnicity group[8]. Findings from England suggest an important role of migrant background in paediatric COVID-19 hospitalisation rates and outcome[9]. The higher prevalence of SARS-CoV-2 antibodies in children with migrant background in the present study resembles these findings. Causes for differences in seroprevalence according to migrant background might be related to temporal and regional variation in incidence, structural and systemic differences such as higher risk occupations, and differences in mobility patterns or households sizes[10]. The volatile interplay of these mechanisms may explain the observed changes of the effect of migrant background on seropositivity over time.

Seroprevalence among adults in Germany was estimated at 14% (as of April 2021)[23]. Preliminary studies in children suggested a somewhat lower seroprevalence in children than in adults[1,6,13].

A gradual increase in testing activity may explain the increasing detection ratio during the pandemic as observed. In March to May 2021, a rate of 2.8 seropositive cases per recalled infection suggests that a higher number of infections in children than adults still went undetected. A large community-based study in Germany identified an underdetection ratio of about 1.8 to 1 for adults of all age groups in the same period[24]. Interestingly, our study revealed that the age of the children influenced the detection rate. The younger the children, the lower the detection ratio, probably due to a lower testing rate and less frequent occurrence of symptoms.

During the COVID-19 pandemic, the rate of respiratory tract infections in children decreased[25], potentially reflecting the effect of contact-restricting measures. Respiratory tract infections were reported more often in seropositive children suggesting that other respiratory infections might have been contained more efficiently.

This study is characterised by a multicentre design and a large sample size. A further strength is information on preceding SARS-CoV-2 test results in a serologic study population, allowing estimation of the proportion of unidentified infections in children in different phases of the pandemic. As the questionnaire did not

specify the timing or applied method of the preceding test (i.e. serologic, PCR or point-of-care/self-administered) and also not the timing of previous tests, the absolute estimate of the under-detection ratios might be biased. With regard to change over time, however, bias will only occur if the number of PCR confirmations per antigen tests changed.

Detectability of ELISA IgG antibodies might not identify all preceding infections. Waning immunity is controversial and an issue in seroprevalence studies. Nevertheless, it should be mentioned that in our data only 0.5% of the seronegative children reported a preceding SARS-2 infection.

Technically, the choice of cut-off values of applied tests are additional critical issues in seroprevalence studies. Our sensitivity analyses gauged the estimator's uncertainty due to different cut-offs, giving a plausible range for seropositivity.

A further potential limitation may pertain to external validity since this is not a population-based study cohort. Unfortunately, we could not validate external validity on the level of the recruiting study sites because we do not have information on the number of admitted children and comparative data regarding recruited and non-recruited patients. However, standardisation for parents' country of origin, age groups, and study site did not account for major changes in the seroprevalence estimates. It is hard to define the catchment areas of the individual study sites. Therefore, we could not account for other determinants such as average socioeconomic status in the catchment area of each hospital in the analysis.

Since the result of the ELISA was not fed back to the participants, we do not assume a bias caused by different willingness to participate due to previous testing. Furthermore, there was no association between the reason for blood sampling or past medical history and the occurrence of SARS-CoV-2 IgG antibodies within the study population (Table 1), suggesting internal validity.

Additionally, we were not able to identify sibling pairs in our study. Thus, we cannot exclude that household clustering might have induced an overestimation of our seroprevalence estimates. However, we would expect this potential effect small and invariant over time, so that this issue should not have affected the overall trend of seroprevalence considerably.

After almost 2 years of pandemic, antibodies against SARS-CoV-2 were not detectable in the majority of children in Germany, which might reflect the effect of differing containment measures. It is currently impossible to determine the individual effects of the measures or other pandemic-influencing determinants on seropositivity. The increase in seroprevalence varied by age group, with a higher prevalence in young children during June to September 2020 and by migrant background. The impact of measures to limit virus spread might have been improved by approaches taking these factors into account. The number of infections still to be expected in children might become a critical challenge for paediatric medical care. The still substantially higher rate of seropositivity despite increasingly testing in schools and day care compared to previously known infections points to the importance of serologic testing to define the risk of outcomes related to SARS-CoV-2 infections in children.

## Methods

**Study design.** The SARS-CoV-2 KIDS study is a hospital-based, multicentre study including cross-sectional seroprevalence surveys of SARS-CoV-2 Immunoglobulin G (IgG) in children (aged ≤ 17 years). In 14 paediatric hospitals across Germany, participants were recruited during their inpatient or outpatient stay, irrespectively of the medical purpose of the stay. Participation involved parental informed consent to use blood samples taken for routine clinical procedures for additional antibody testing against SARS-CoV-2. Children with corrected gestational age less than 37 completed weeks, severe congenital or acquired immune deficiencies, immunosuppression due to chemotherapy or stem cell transplantation, treatment due to life-threatening emergencies, and children already vaccinated against SARS-CoV-2 were excluded from participation. Repeated participation was not possible.

Additionally, an anonymous parental questionnaire was deployed to obtain demographic and clinical information. The English version of the questionnaire is presented in the Supplementary Information.

All blood samples were tested at the routine diagnostic department of the Charité Medical Centre, Berlin.

A unique, anonymous identifier variable was used to link serum samples and questionnaires.

**Detection of antibodies.** We used a commercially available anti-SARS-CoV-2 Enzyme-Linked Immunosorbent Assay (ELISA - Euroimmun Medizinische Diagnostika AG, Lübeck, Germany) to detect IgG specific for the S1 domain of SARS-CoV-2 spike protein[26]. The batches of the ELISA are listed in the Supplementary Information. Briefly, serum samples were analysed at a 1:101 dilution using the automated EUROLabWorkstation ELISA platform. The ELISA yields an optical density (OD) ratio, the quotient of OD in a sample and OD of a calibrator tested in parallel, providing a semi-quantitative measure for antibodies in serum sample. We considered samples with an OD ratio above 1.1 as Anti-SARS-CoV-2 IgG positive.

We tested for neutralising antibodies by Plaque Reduction Neutralization Tests (PRNT) on a subsample of 252 sera, selected to cover the whole range of ELISA OD ratios. PRNTs were performed as previously described[27,28]. The lowest tested serum dilution in log2-dilution series was 1:10. Serum dilutions causing plaque reductions of 90% (PRNT-90) and 50% (PRNT-50) were recorded as titres, and sera showing PRNT-50 ≥ 1:20 were classified as positive for neutralising antibodies.

We evaluated the manufacturer-recommended ELISA threshold by using Receiver Operating Characteristic (ROC) analyses based on data of the presence of neutralising antibodies and recalled infection status.

Technical details for the ELISA, PRNT, and the statistical approach for evaluating the ELISA threshold are summarised in the Supplementary Methods.

**Differences in seroprevalence by sample characteristics.** We defined three categories of migrant background according to whether a foreign country of origin was reported for both, one or none of the parents. If information on only one parent was provided, the child was assigned according to this information as having both or no parents from abroad.

Age was classified as children aged under three, 3–11-year old, and 12 years or older, assuming that children of different ages are exposed to the virus to varying degrees.

For each variable of interest, we calculated point estimates for the prevalence of SARS-CoV-2 antibodies and 95% confidence intervals (95% CI) based on individual Wald tests. Pearson chi-square tests using Bonferroni correction were used for the comparison of average seroprevalence by sample characteristics.

**Assessment of the temporal course of seroprevalence.** To investigate changes in seroprevalence over time, we fitted a logistic regression model and plotted the predicted probability of seropositivity depending on time modelled as a non-linear b-spline function. Additionally, following a variable selection based on a priori hypotheses and significant associations with seropositivity, we adjusted this model for the potential confounders migrant background and age group and assessed interactions of these two variables with time of antibody testing. In order to visualise the temporal trends of seroprevalence by age and migrant background, we stratified the logistic regression model according to each of these two variables and plotted the resulting b-spline curves of seropositivity, respectively. Further, we assessed the seroprevalence estimates per age group in three periods of the pandemic: June 2020 to September 2020 with low nation-wide numbers of COVID-19 positive patients per day, October 2020 to February 2021 with the highest number of COVID-19 patients and strict restrictive measures, and March to May 2021.

In order to minimise potential differences with the overall population of children in Germany, two-month prevalence estimates were standardised for migrant background (at least one parent with a country of origin outside Germany compared to none) and age groups, taking the distribution at the respective study centre into account. We applied the direct method using the micro census data 2019[29] and age structure population data provided by the German Federal Statistical Office[30].

**Associations with previous SARS-CoV-2 testing.** We assessed whether participants, who reported to be previously tested differ from participants without being tested previously by comparing patient characteristics using Pearson chi-square tests and reported Bonferroni corrected p-values.

In order to identify changes of the detection ratio during the observation period, we considered a SARS-CoV-2 infection suggested by seropositivity as 'known' in case of a recalled previous positive test on SARS-CoV-2. We estimated the detection rate, defined as the ratio of children with positive ELISA and 'known'

previous infection to all ELISA positive children, by age group and for the three previously mentioned periods of the pandemic.

The significance level was set at 5%. Statistics were calculated using SAS, version 9.4 (SAS Institute Inc., Cary, NC, USA) or R 3.5.1 (R Foundation for Statistical Computing, Wien, Austria).

Initial ethical approval was obtained from the Ethics Committee of the Medical Faculty of the Heidelberg University (No. 2020-536N). Ethics committees of the other study centres subsequently also independently approved the study protocol (Ethics Committee of the Medical Faculty of the Ludwig-Maximilians-University Munich No. 20–348, Charité Berlin, Technical University Dresden No. BO-EK142042020, Medical Faculty of the HHU Düsseldorf No. 2020-936, Saarland Medical Association No. 65/20, Hamburg Medical Association No. MC-142/20, Nordrhein Medical Association No. 2020099, Albert-Ludwigs-University freibug No. 243/20, Medical Faculty of the RWTH Aachen No. 081/20, Medical Faculty of the Justus-Liebig University Giessen No. 61/20, MMH Hannover No. 9041_BO_K_2020, Julius-Maximilians University Würzburg No. 92/20_z, Technical University Munich No. 264/20S). All parents/guardians gave written informed consent and children assented to the participation when appropriate for their age.

**Reporting summary**. Further information on research design is available in the Nature Research Reporting Summary linked to this article.

## Data availability
The raw data shown in the manuscript are subject to controlled access because they are the subject of ongoing work and will be made available on request to the corresponding author. Source data are provided with this paper.

## Code availability
The code used to generate the presented results is available online under https://osf.io/am2ck/ https://doi.org/10.17605/OSF.IO/AM2CK).[31]

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

## Acknowledgements
The work was supported by funding from the German Federal Ministry of Education and Research, BMBF (FKZ: 01KI20131A). The sponsor was not involved in the design and conduct of the study; collection, management, analysis, and interpretation of the data; preparation, review, or approval of the manuscript; and not in the decision to submit the manuscript for publication. We thank all parents and patients for participation in the study. Institute of Social Paediatrics and Adolescent Medicine, Division of Paediatric Epidemiology, Ludwig-Maximilians-University Munich, Germany: Selina Becht, Veronika Kaiser, Tamara Weindl (Digitisation and validation of questionnaires, support of study coordination, and communication). Paediatric Infectious Diseases, Department of Paediatrics, Medical Faculty Mannheim, Heidelberg University, Mannheim, Germany: Dr. Michael Eichinger (reading the manuscript, constructive suggestions); Giselle Decker

(sera sample collection); Andrea Hilpert (sera sample collection); Claudia Fahandezh-Saadi (technical assistance). Department of Paediatric Respiratory Medicine, Immunology, and Critical Care Medicine, Charité Universitätsmedizin Berlin, Berlin, Germany: Josefine Dobbertin-Welsch, Jonathan Groß, Sveva Castelli, Inga Tometten, Mehrak Yosefi, Songül Yürek (sera sample collection). Department of Paediatrics, University Hospital, and Medical Faculty Carl Gustav Carus, Technische Universität (TU) Dresden, Dresden, Germany: Christiane Walther and Caty Ullmann (sera sample collection). Universities of Giessen and Marburg Lung Centre, German Centre of Lung Research, Department of Paediatrics, Justus-Liebig-University Giessen, Giessen, Germany: Prof. Dr. Klaus-Peter Zimmer for his support of the study and all colleagues of the Department of Paediatrics, especially Platonas Karatsiolis, Oezlem Satirer and Andrea Kessel for sera sample collection; Mathilda Dana Bach, Janine Lea Glaser and Luisa Gersmann for the assistance of the study. University Children's Hospital, University Medical Centre Hamburg- Eppendorf, Hamburg, Germany: Jan-Philipp Heinrich, Marcus Kania, Yella Woo. Department of General Paediatrics, Neonatology, Children's Hospital Medical Centre, Saarland University Hospital, Homburg/Saar, Germany: Prof. Dr. Michael Zemlin and Prof. Dr. Arne Simon for their support of the study; Tabea Reinhardt and Tilman Rohrer for the assistance of the study. Helios Klinikum Krefeld, Krefeld, Germany: Dr. Andreas Christaras. Dr von Hauner Children's Hospital, University Hospital, Ludwig-Maximilians-University, Munich, Germany: Laura Kolberg (study coordination).

## Author contributions

A.-L.S. had full access to all the data in the study and takes responsibility for the integrity of the data and the accuracy of the data analysis. R.v.K., H.S., A.-L.S: conceptualisation, study design, funding acquisition, data collection, data analysis, data interpretation, visualising, writing. C.D., V.C., L.B.: funding acquisition, data collection, data analysis, data interpretation, writing. I.S.: funding acquisition, project administration. M.J; A.G.; C.S.: literature search, study design, data collection, study management at the respective study sites, patient recruitment, data acquisition, writing—review & editing. A.B.: data analysis, methodology, writing—review & editing. E.V.; N.W.; H.v.B.; F.L.; K.W.; M.H.; U.S.; C.-M.C.; L.N.; A.C.M.; U.S.-S.; G.H.; M.W.; A.M.J.; T.N.; S.F.-O.; U.v.B.; J.H.; U.B.; J.G.L.: study management at the respective study sites, patient recruitment, data acquisition, interpreting of data, writing—review & editing

## Funding

## Competing interests

The study centres received academic research funding from the Federal Ministry of Education and Research (BMBF) for study planning, study management and reimbursement for the assay of this study. V.C. reported a patent on methods and reagents for diagnosis of SARS-CoV-2 infection (Pub number 20210190797) of Charité and Euroimmun GmbH.H.v.B. reported receiving honoraria for ongoing reports on the safety profile of an IgG-product from Octapharma GmBH, honoraria for lectures from CSL Behring, honoraria for advisory boards from Takeda and Swedish Orphan Biovitrum AB (publ) (SobiTM) and the 'treatment guideline for Primary 'immunodeficiencies' is under his unpaid leadership. L.N. reported institutional fees for study participation by the German Centre for Lung Research, Vertex Pharmaceuticals and Boehringer Ingelheim. He participates on Trial Steering Committee for CF STORM and is the medical leader of the German CF-registry, and the pharmacovigilance study manager of the ECFSPR.G.H. reported receiving consulting fees from Sanofi GmbH and honoraria for lectures from MedUpdate and Abbvie.No other disclosures were reported.
