## [Peer Review File · Nature Communications]

Cross-sectional seroprevalence surveys of SARS-CoV-2 antibodies in children in Germany, June 2020 to May 2021REVIEWER COMMENTS

Reviewer #1 (Remarks to the Author):

The authors present data from a multicentre study in Germany where more than 10,000 children from 14 study centers were tested for SARS-Cov2-antibodies. The data are interesting however the study design is – necessarily – not optimal as the authors acknowledge since it is not a population based study. It cannot be excluded that the reason for attending the hospital is linked with a current or previous infection. The reviewer is however aware that a population based study is very difficult to perform.

The description of the statistical procedures is insufficient and the methods used appear to be sub-optimal. I understand that individual data are available however this has not been made explicit. According to the last para in the introduction, the main aim of the study is to assess the temporal course of seroprevalence of SARS-Cov2-antibodies in German children aged up to 17 years during the period June 2020 to May 2021, and to relate it to age, migrant background and prior infections. For this aim, a logistic regression analysis would be a better method which would allow to simultaneously investigate the effect of age, migrant background, medical purpose of hospital visit and time on the outcome. The method using splines to describe the time course is appropriate and this could be incorporated in the logistic model. Given the large sample size, it would also allow to some degree to investigate interaction between the above variables.

The authors did not present data by study center. I wonder whether the prevalences differ between centers. At least a possible heterogeneity between centers should be taken into account. Migrant background has been defined in this study according to the official definition of the federal statistical office. However, this definition has limited usefulness since, for example, it does not distinguish between children of refugees, children of parents from, say, Turkey, and children where one parent is from a neighbouring country, say Austria. Some more distinction would be useful in this study, for example (i) both parents from abroad (ii) one parent from abroad and (iii) both parents without migration background

Discussion: "A recent seroprevalence study in Bavaria, a federal state of Germany, found seroprevalence estimates in 1-5 and 6-10 year old children of 5.6% and 8.4% in February 2021, respectively.¹⁰ The corresponding estimates in our study were 9.8% and 7.8%. There seems to be a higher seroprevalence in young children in the present study. Differences in the utilisation of medical services (hospital versus private offices), study design, and method of antibody testing could contribute to this finding." This remark is not very helpful. First, method of antibody testing on one hand and study design/differences in utilisation on the other hand are quite different issues. How could the method of antibody testing explain that in one study the rate goes up and in the other study it goes down?

Reviewer #2 (Remarks to the Author):

Dear authors,

You investigated more than 10000 sera of children aged <1 to 17 years from 14 paediatric hospitals in Germany collected between June 2020 to May 2021 by Euroimmun ELISA. Seroprevalences increased from 2.0% in summer 2020 to 10.8% in March 2021 and did not further increase during the two months afterwards. There were higher seroprevalences in younger children in the first investigation period which could be explained by containment measures (reducing the contacts of older children) and closer contacts of the youngest children to adults during household transmissions. Moreover, seroprevalences in children with migrant background were higher than in those without which was explained by possible differences in mobility patterns, in household sizes and other factors.

A subset of 252 samples was investigated by plaque reduction neutralisation assay on Vero E6 cells. None of the sera with Euroimmun ELISA OD ratio below 0.8 was tested positive for neutralising antibodies. The majority of sera with OD ratio > 1.1 had neutralising activity.

With respect to the further development during the pandemic which reflect increasing infection

rates of children during the 4th wave in Germany the results of the study are of great importance. The study is well-designed, the methodology is sound and meets the standards. The discussion is profound. The data support the conclusions. The Supplement provides enough details on the methods.

Some remarks:

Abstract: Please include the results regarding migration background in the abstract.

Line 114 between "... years)." and "In" a gap is lacking

Line 172 the migrant background should be explained in more depths. Table 1 and the study specific questionnaire show that the country of origin of the parent and the language spoken in the family were recorded. Please define the migration background in the Study design (lines 112 and following).

Supplement: The batches of the SARS-CoB-1-S1 ELISA used should be provided.

Reviewer #3 (Remarks to the Author):

This is an interesting and timely paper addressing an important public health concern at the moment: the seroprevalence and determinants of SARS-CoV-2 antibodies in children. Currently, data in this regard is sparse and the existing studies are hampered by relatively small sample sizes. The present study (SARS-CoV-2 KIDS study) employs a hospital-based approach assessing seroprevalence in a large population (>10,358) of children (defined as ≤ 17 years) in 14 pediatric hospitals across Germany. Although interesting, I do have a number of concerns that should be addressed before the paper would be suitable for publication:

Abstract:

- That there was "... little change after March 2021..." , could also simply be due to the fact that the serosurvey ended only two months later, i.e. in May 2021.
- " Seropositive cases per recalled infection increased from one in 8.6 to one in 2.8". I think this should be: " Seropositive cases per recalled infection increased from 2.8 to 8.6".

Introduction:

- The authors specifically mention the contribution of a migrant background as a potentially important, but understudied determinant of SARS-CoV-2 infection. However, the introduction is lacking a clear rationale / hypothesis in this regard.

Methods:

- The seroprevalence estimates do not seem to be adjusted for household and geographic clustering of cases; would it be possible to take this into account?
- No information is provided regarding the type (i.e. serologic, PCR or point-of-care / self-administered) and timing of the previous tests. Would it be possible to obtain some more information in this regard, at least in a subset of the participants?
- Were there differences in participant characteristics between those who had previous test results and those who did not?

Results:

- The study population: although the authors provide information on the study population, detailed information regarding the source population is lacking: how many children were seen in total in the 14 hospitals during the serosurvey period? What was the response rate? Were there any differences in characteristics between the people who were included and who were not included in the serosurvey? This issue is particularly important to allow for assessment of potential selection bias.
- Could the differences in seroprevalence rates between children with and without a migration background be explained by socioeconomic factors, i.e. do the differences in seroprevalence rates

remain after adjustment for, e.g., socioeconomic status of the parents? I appreciate that this information might not be available on an individual level (at least, this appears not to be included in the questionnaire in the supplement), but as a proxy the authors might consider using the average socioeconomic status in the catchment area of each hospital.

- Lines 189-191: "... Of all seropositive children, 22.6% (n=96/424) had one or more respiratory tract infections with symptoms such as fever or shortness of breath, as opposed to 13.9% in children without SARS-CoV-2 antibodies (n=1336/9563, $p < 0.0001$)...". What about other symptoms? The questionnaire in the supplement also contains questions on other symptoms.

- Figure 2: Please include a (supplementary) figure with spline regression lines for each of the three age categories separately.

- Figure 3: Helpful to put the absolute numbers in (or below) the bar graphs for each age category.

- Table 1:

o The authors include "Reason for hospitalization" as a variable (with six categories) which is compared between the seropositive and seronegative children. However, in addition, I would suggest to elaborate more on the different departments to which the children were admitted as this would allow for a better comparison between the different pre-existing conditions as risk factors for seropositivity. For example, were children with pre-existing hematological / immunological conditions more likely to have tested positive for SARS-CoV2 compared to other? Similarly, were children with obesity-related (pre-)conditions more likely to become seropositive?

o Is there any information regarding the type (i.e. serologic, PCR or point-of-care / self-administered) and timing of the previous tests?

o Was the group with a previous test available different from the group with no previous test results available? If yes, in what respect?

- Table 2:

o Could this table be expanded by including stratified prevalence rates for the three different age groups?

Discussion:

- The authors speculate about the various potential causes of the differences in the temporal patterns of seropositivity seen between the three different age strata, especially regarding the potential influence of different national containment strategies. For a clearer appreciation of these relations, it would be helpful to have a graphical overview of the temporal seropositivity estimates in conjunction with the national containment rules.

Supplementary Material:

- Line 502: page number in the index missing.

- Lines 574-577: " (3) At the manufacturer-recommended threshold of 1.1, sensitivity and specificity to predict a PRNT-50 ≥ 1 : 20 were 94.4% (95% CI 84.6, 98.8) and 41.6% (95% CI 32.9, 50.8), respectively. The manufacturer's threshold predicted infection status per questionnaire with sensitivity of 71.3% (95% CI 63.8%, 78.0%) and specificity of 95.6 (95% CI 94.9, 96.2)". ==> It is unclear what the sensitivity of 71.3% and specificity of 95.6% refer to; is the answer to the questionnaire regarding a positive test used as the "gold standard"? This would be erroneous.

- Lines 586-592:

The authors state that: "... The temporal trends of seroprevalence throughout the study period (eFigure 3) and the predicted probability of a positive serum test according to b-spline regression models (eFigure 4) were similar for all three tested thresholds, confirming the robustness of the applied ELISA test. Therefore, we assume that any cut-off within the range of OD ratios of 0.61 and 2.04 delivers an acceptable estimate of seroprevalence trend. Based on these observations, we consider the manufacturer-recommended threshold at OD ratio 1.1 a valid and practical classifier to conduct serosurveys in children, because it permits comparison with adult serosurvey using the same system." ==> I have trouble following this line of reasoning: although I understand that using different OD ratios does not impact the overall seroprevalence trend, it of course does affect the absolute estimates of the seroprevalence. This should be explained better.

Resubmission

SARS-CoV-2 Antibodies in Children: a one-year seroprevalence study from June 2020 to May 2021 in Germany

manuscript NCOMMS-21-49166

Munich, 2/24/2022

Dear Reviewers,

We are grateful for the opportunity to revise the manuscript and tables/figures in consideration of your helpful comments. We gladly implemented all your comments. Implementation required some restructuring of the paper. Track changes was used to highlight the alterations in the revised version.

In the following our replies to the your comments.

Reviewer #1:

The authors present data from a multicentre study in Germany where more than 10,000 children from 14 study centers were tested for SARS-Cov2-antibodies. The data are interesting however the study design is – necessarily – not optimal as the authors acknowledge since it is not a population based study. It cannot be excluded that the reason for attending the hospital is linked with a current or previous infection. The reviewer is however aware that a population based study is very difficult to perform.

Reply: Thank you for the comment. Since we were able to include a large sample size over a one-year observation period, we feel that our findings are useful despite limitations regarding population representativeness.

The description of the statistical procedures is insufficient and the methods used appear to be sub-optimal. I understand that individual data are available however this has not been made explicit. According to the last para in the introduction, the main aim of the study is to assess the temporal course of seroprevalence of SARS-Cov2-antibodies in German children aged up to 17 years during the period June 2020 to May 2021, and to relate it to age, migrant background and prior infections. For this aim, a logistic regression analysis would be a better method which would allow to simultaneously investigate the effect of age, migrant background, medical purpose of hospital visit and time on the outcome. The method using splines to describe the time course is appropriate and this could be incorporated in the logistic model. Given the

large sample size, it would also allow to some degree to investigate interaction between the above variables.

Reply: We thank the reviewer for the idea of using a multivariable logistic regression model, which we gladly implemented.

We adjusted our time-course model for those variables for which we considered as potential confounders, i.e. which were associated with seropositivity, but were not in the causal pathway between time of antibody test and seropositivity. As shown in Table 1, we observed significant associations with seropositivity after Bonferroni correction for age group, parental country of origin and language spoken in the family. As the latter two variables were obviously collinear, we decided to put country of origin into the model and leave the language variable out. The variables ‘Test of SARS-CoV-2 infection in the past’, ‘History of respiratory diseases since March 2020’, ‘History of pneumonia since March 2020’ and ‘History of hospitalisation due to pneumonia since March 2020’ were likely to be in the causal pathway of seropositivity and time of antibody test and were therefore not included in the model. In addition, we investigated interaction terms between migration background and time of antibody test, as well as between age groups and time of antibody test.

The results section of the manuscript was revised accordingly. We report the odds ratios of the variables in the multivariable logistic model and added a comparison of the partial residuals over time in the raw and adjusted model in a new eFigure 4.

The authors did not present data by study center. I wonder whether the prevalences differ between centers. At least a possible heterogeneity between centers should be taken into account

Reply: We have added a supplementary figure (eFigure 1) showing seroprevalence in the different phases of the pandemic separately by study centre. As expected, there was some heterogeneity between the study centres as the infection numbers and the course of the pandemic also differed between the regions. Since the objective of the study was to provide nationwide temporal trends we had not pursued this issue in first instance.

Migrant background has been defined in this study according to the official definition of the federal statistical office. However, this definition has limited usefulness since, for example, it does not distinguish between children of refugees, children of parents from, say, Turkey, and children where one parent is from a neighbouring country, say Austria. Some more distinction would be useful in this study, for example (i) both parents from abroad (ii) one parent from abroad and (iii) both parents without migration background

Reply: Thank you for this comment. As suggested, we have now classified migrant background into the three subclasses and found that, interestingly, the prevalence was highest when both parents came from abroad. We have adapted the text, figures and tables accordingly.

Discussion: "A recent seroprevalence study in Bavaria, a federal state of Germany, found seroprevalence estimates in 1-5 and 6-10 year old children of 5.6% and 8.4% in February 2021, respectively.¹⁰ The corresponding estimates in our study were 9.8% and 7.8%. There seems to be a higher seroprevalence in young children in the present study. Differences in the utilisation of medical services (hospital versus private offices), study design, and method of antibody testing could contribute to this finding." This remark is not very helpful. First, method of antibody testing on one hand and study design/differences in utilisation on the other hand are quite different issues. How could the method of antibody testing explain that in one study the rate goes up and in the other study it goes down?

Reply: Thank you much for pointing to this implausibility. Indeed, a difference in antibody testing cannot explain differences in one age group only. We omitted this argument accordingly.

Reviewer #2:

Dear authors,

You investigated more than 10000 sera of children aged <1 to 17 years from 14 paediatric hospitals in Germany collected between June 2020 to May 2021 by Euroimmun ELISA. Seroprevalences increased from 2.0% in summer 2020 to 10.8% in March 2021 and did not further increase during the two months afterwards. There were higher seroprevalences in younger children in the first investigation period which could be explained by containment measures (reducing the contacts of older children) and closer contacts of the youngest children to adults during household transmissions. Moreover, seroprevalences in children with migrant background were higher than in those without which was explained by possible differences in mobility patterns, in household sizes and other factors.

A subset of 252 samples was investigated by plaque reduction neutralisation assay on Vero E6 cells. None of the sera with Euroimmun ELISA OD ratio below 0.8 was tested positive for neutralising antibodies. The majority of sera with OD ratio > 1.1 had neutralising activity.

With respect to the further development during the pandemic which reflect increasing infection rates of children during the 4th wave in Germany the results of the study are of great importance. The study is well-designed, the methodology is sound and meets the standards. The discussion is profound. The data support the conclusions.

The Supplement provides enough details on the methods.

Some remarks:

Abstract: Please include the results regarding migration background in the abstract.

Reply: We added the following sentence to the abstract '*Rates increased by parent migrant background (2.8 (German), 4.4 and 7.8 for one and two parents born abroad.)*'. We hope this meets your request.

Line 114 between "... years)." and "In" a gap is lacking

Reply: Thank you for reading carefully. We have corrected it.

Line 172 the migrant background should be explained in more depths. Table 1 and the study specific questionnaire show that the country of origin of the parent and the language spoken in the family were recorded. Please define the migration background in the Study design (lines 112 and following).

Reply: In reply to reviewer 1's comment, we have redefined migrant background using 3 categories: Participants with both parents from Germany, participants with one parent with a foreign country of origin, and participants with both parents from abroad. The method section now describes how the variable migrant background is defined (not in part 'Study design' but in part 'Statistical analysis'). We hope that the changes made at the appropriate passages in the manuscript make the point easier to understand. Since language spoken in the family is not part of the common definition of migrant background we analysed it separately. The strength of the association was similar. We used country of origin of the parents to avoid collinearity.

Supplement: The batches of the SARS-CoB-1-S1 ELISA used should be provided.

Reply: Thanks for this idea. We included the batches in the Supplement of the manuscript.

Reviewer #3:

This is an interesting and timely paper addressing an important public health concern at the moment: the seroprevalence and determinants of SARS-CoV-2 antibodies in children. Currently, data in this regard is sparse and the existing studies are hampered by relatively small sample sizes. The present study (SARS-CoV-2 KIDS study) employs a hospital-based approach assessing seroprevalence in a large population (>10,358) of children (defined as ≤ 17 years) in 14 pediatric hospitals across Germany. Although interesting, I do have a number of concerns that should be addressed before the paper would be suitable for publication:

Abstract:

- That there was "... little change after March 2021...", could also simply be due to the fact that the serosurvey ended only two months later, i.e. in May 2021.

Reply: Thank you. Indeed we censored the time of observation at the end of May 2021. The correct wording would be 'up to May 2021'. Although not included in this analysis, we were able to maintain the serosurvey beyond May 2021. The data from June 2021 to October 2021 will be analysed and published separately.

- "Seropositive cases per recalled infection increased from one in 8.6 to one in 2.8 ". I think this should be: "Seropositive cases per recalled infection increased from 2.8 to 8.6 ".

Reply: A simple description of these interdependencies is complex. We appreciate your proposal for the wording, but it should be 'decreased': 'Among the seropositive cases, seropositive cases per recalled infection decreased from 8.6 in June to September 2020 to 2.8 in March to May 2021'.

Introduction:

- The authors specifically mention the contribution of a migrant background as a potentially important, but understudied determinant of SARS-CoV-2 infection. However, the introduction is lacking a clear rationale / hypothesis in this regard.

Reply: Thanks for this comment. We added a sentence to the introduction explaining why studying migrant background in our analysis is interesting.

Methods:

- The seroprevalence estimates do not seem to be adjusted for household and geographic clustering of cases; would it be possible to take this into account?

Reply: Thank you, for raising this potentially important issue. Unfortunately, our questionnaire does not allow us to identify siblings. Therefore we added this point to the limitations: 'Additionally, we were not able to identify sibling pairs in our study. Thus, we cannot exclude that household clustering might have induced an overestimation of our seroprevalence estimates. However, we would expect this potential effect small and invariant over time, so that this issue should not have affected the overall trend of seroprevalence considerably.'

Geographical clustering was taken into account by integrating the respective study centres into standardising the two-month seroprevalence estimates (eTable 2).

- No information is provided regarding the type (i.e. serologic, PCR or point-of-care / self-administered) and timing of the previous tests. Would it be possible to obtain some more information in this regard, at least in a subset of the participants?

Reply: We agree that this would be helpful information. When the study design was planned in April 2020, there were no different types of tests, so this explicit question was unfortunately not integrated into the questionnaire. Due to the anonymous study design, it is unfortunately also not possible to obtain this information retrospectively, not even for a subset.

- Were there differences in participant characteristics between those who had previous test results and those who did not?

Reply: Thank you for raising this interesting question. The study design excluded feedback of the ELISA test results to the participants to exclude previous testing as a potential incentive for study participation. For clarification, we have added the following sentence: *'Since the result of the ELISA was not fed back to the participants, we do not assume a bias caused by different willingness to participate due to previous testing.'*

In another supplementary table (eTable 1), we present the comparison of the characteristics of participants who reported to have been tested in the past with those who reported not to have been tested previously. As expected, tested participants more often reported a history of respiratory disease, pneumonia or hospitalisation due to pneumonia. Also, the proportion of children with previous illnesses was somewhat higher in this group, presumably due to a higher testing rate in children with heart diseases, haematological/oncological disease, neurological/neuromuscular diseases, gastrointestinal or chronic renal diseases.

Results:

- The study population: although the authors provide information on the study population, detailed information regarding the source population is lacking: how many children were seen in total in the 14 hospitals during the serosurvey period? What was the response rate? Were there any differences in characteristics between the people who were included and who were not included in the serosurvey? This issue is particularly important to allow for assessment of potential selection bias.

Reply: This is an important issue. Indeed, it would have been ideal to know more about families refusing participation or not being asked to participate. Unfortunately, we do not have any data on non-participants. Inclusion was driven by physicians asking families for their consent and willingness to participate. Selection bias might arise if selection or participation was associated with difference in interests in the test results. To minimize such bias we designed the study without feedback of the test result to the participants.

- Could the differences in seroprevalence rates between children with and without a migration background be explained by socioeconomic factors, i.e. do the differences in seroprevalence rates remain after adjustment for, e.g., socioeconomic status of the parents? I appreciate that this information might not be available on an individual level (at least, this appears not to be included in the questionnaire in the Supplement), but as a proxy the authors might consider using the average socioeconomic status in the catchment area of each hospital.

Reply: This is an interesting proposal. We are convinced that migration background is a surrogate for other seroprevalence-increasing effects. However, unfortunately, we can only speculate about

the actual causes. It is likely that higher-risk occupations of the parents, differences in mobility patterns or households sizes, but also, as you suggested, family's socioeconomic status may have an influence.

Unfortunately, the catchment areas of the participating hospitals do not overlap with specific municipalities or districts for which area-level deprivation indices would be available. Although it might be interesting to identify whether the effect of migrant background differs by socioeconomic status, this cannot be addressed in these data and average data as surrogate were not available.

- Lines 189-191: "... Of all seropositive children, 22.6% (n=96/424) had one or more respiratory tract infections with symptoms such as fever or shortness of breath, as opposed to 13.9% in children without SARS-CoV-2 antibodies (n=1336/9563, p=<0.0001)...". What about other symptoms? The questionnaire in the Supplement also contains questions on other symptoms.

Reply: We apologise for being imprecise about that. The presented results refer to question *„How many respiratory infections with the symptoms of fever and shortness of breath has your child gone through since 1st March 2020 until today? ‘*. The query of the other symptoms (skin rash, conjunctivitis, hypotension, shock, features of myocardial abnormalities, coagulopathy, or acute gastrointestinal Problems) refer to symptoms of PIMS. The enquiry into these symptoms aims to investigate whether seropositive children are more likely to report signs suggestive of PIMS than seronegative children. This question, however, will be addressed in a separate analysis.

- Figure 2: Please include a (supplementary) figure with spline regression lines for each of the three age categories separately.

Reply: A figure (eFigure 3) displaying spline regression lines for each of the three age categories separately is now integrated in the Supplement.

- Figure 3: Helpful to put the absolute numbers in (or below) the bar graphs for each age category.

Reply: That is a good hint. We have integrated the point estimates and the absolute numbers of the individual categories in the figure.

- Table 1:

o The authors include "Reason for hospitalisation" as a variable (with six categories) which is compared between the seropositive and seronegative children. However, in addition, I would suggest to elaborate more on the different departments to which the children were admitted as this would allow for a better comparison between the different pre-existing conditions as risk factors for seropositivity. For example, were children with pre-existing hematological / immunological conditions more likely to have tested

positive for SARS-CoV2 compared to other? Similarly, were children with obesity-related (pre-)conditions more likely to become seropositive?

Reply: Unfortunately, we do not know the departments the children were admitted to, as this was not asked in the questionnaire. However, we do have information about the children's pre-existing diseases (which does not necessarily correspond to the reason for the hospital stay). Nevertheless, we agree with your suggestion and have selected some pre-existing diseases and compared them between the seropositive and seronegative children. See the results attached in Table 1. After adjustment for multiple testing, no pre-existing disease was significantly associated with seropositivity.

o Is there any information regarding the type (i.e. serologic, PCR or point-of-care / self-administered) and timing of the previous tests?

Reply: Unfortunately, there is no information about the type and timing of the previous tests. At the time of study design, a PCR test was the only test available in Germany. Therefore, the questionnaire did not include a specification of the test types. However, since we assume that PCR confirms all point-of-care or self-administered tests in Germany, all reported previous positive test results are likely PCR-confirmed SARS-CoV-2 infections.

o Was the group with a previous test available different from the group with no previous test results available? If yes, in what respect?

Reply: We feel that this question has already been addressed previously (see the reply to your question on page 7) and hope that the reply to your previous question clarifies this issue.

- Table 2:

o Could this table be expanded by including stratified prevalence rates for the three different age groups?

Reply: Thank you very much for this nice idea, which we have gladly implemented. It is evident that in the group of the youngest children, the proportion of seropositive children with a previously positive test result in relation to all seropositive children in this age group is lower than in the other two age groups. However, the decreasing trend of this proportion over the different phases of the pandemic is clearly visible in all three age groups. Nevertheless, the detection rate for the under-threes of 6 seropositive cases per recalled infection remains lower than for the other two age groups with a detection rate of 2 resp. 3 seropositive cases per recalled infection.

The result was also integrated into the description of the results.

Discussion:

- The authors speculate about the various potential causes of the differences in the temporal patterns of seropositivity seen between the three different age strata, especially regarding the potential influence of different national containment strategies. For a clearer appreciation of these relations, it would be helpful to have a graphical overview of the temporal seropositivity estimates in conjunction with the national containment rules.

Reply: We would be happy to provide this data in detail, but Germany has no unique national containment strategy. Instead, there were different containment strategies over time in each of the 16 federal states of Germany, and thus it would be impossible to depict them.

Supplementary Material:

- Line 502: page number in the index missing.

Reply: We apologise for this error and have corrected it.

- Lines 574-577: "(3) At the manufacturer-recommended threshold of 1.1, sensitivity and specificity to predict a PRNT-50 $\geq 1 : 20$ were 94.4% (95% CI 84.6, 98.8) and 41.6% (95% CI 32.9, 50.8), respectively. The manufacturer's threshold predicted infection status per questionnaire with sensitivity of 71.3% (95% CI 63.8%, 78.0%) and specificity of 95.6 (95% CI 94.9, 96.2)". ==> It is unclear what the sensitivity of 71.3% and specificity of 95.6% refer to; is the answer to the questionnaire regarding a positive test used as the "gold standard"? This would be erroneous.

Reply: Thank you for pointing to this important issue. We are well aware that we did not use the 'gold standard': this would require serological testing following a PCR test. Our intention for this analysis was to get more information about the validity of our results using the best available information we had. We point to this limitation in the Supplement: *Sensitivity and specificity with reference to preceding SARS-CoV-2 infection based on recalled previous positive test results in the questionnaire refer to 'a best available arbitrary standard' and not to a 'gold standard', which must be taken into account when interpreting the values.*

- Lines 586-592:

The authors state that: "... The temporal trends of seroprevalence throughout the study period (eFigure 3) and the predicted probability of a positive serum test according to b-spline regression models (eFigure 4) were similar for all three tested thresholds, confirming the robustness of the applied ELISA test. Therefore, we assume that any cut-off within the range of OD ratios of 0.61 and 2.04 delivers an acceptable estimate of seroprevalence trend. Based on these observations, we consider the manufacturer-recommended threshold at OD ratio 1.1 a valid and practical classifier to conduct serosurveys in children, because it permits comparison with adult serosurvey using the same system.". ==> I have trouble following this line of reasoning: although I understand that using different OD ratios does not impact the

overall seroprevalence trend, it of course does affect the absolute estimates of the seroprevalence. This should be explained better.

Reply: We agree with you that this was not described clearly enough at this point. We have changed the text and hope the description is now more understandable. We added '*Despite similar trends, it must be acknowledged that the seroprevalence estimates at any point of the observation period would be substantially shifted depending on the chosen cut-off value. The higher the cut-off, the lower the seroprevalence estimates. The strength of this study design is to depict the temporal course and to suggest a reasonable range for the seroprevalence at a given point in time.*'

Thank you very much for your support!

We hope that the manuscript, in its present form, is acceptable for publication in nature communications as a research article.

Respectfully,

Rüdiger von Kries,

for all authors

REVIEWERS' COMMENTS

Reviewer #1 (Remarks to the Author):

All my comments have carefully been taken into account. I do not have further comments.

Reviewer #3 (Remarks to the Author):

The authors have adequately addressed my previous comments. I only suggest the following:

In response to several of my previous comments regarding

- 1) type (i.e. serologic, PCR or point-of-care / self-administered) and timing of previous tests,
- 2) the study population (i.e. how many children were seen in total in the 14 hospitals during the serosurvey period? What was the response rate? Were there any differences in characteristics between the people who were included and who were not included in the serosurvey?), and
- 3) source population / catchment area

The authors state that this information is / was not available to them. I understand the challenges faced by the authors to collect this information, but would recommend to include and discuss the lack of information in these respects as further potential limitations of the study.

Resubmission

SARS-CoV-2 Antibodies in Children: a one-year seroprevalence study from June 2020 to May 2021 in Germany

Title changed to:

Cross-sectional seroprevalence surveys of SARS-CoV-2 antibodies in children in Germany, June 2020 to May 2021

manuscript NCOMMS-21-49166A

Munich, 3/31/2022

Dear Reviewers,

Thank you for your time and effort in reviewing our manuscript a second time.

In the following you can find the responses and changes we made to the manuscript based on your new comments. Track changes was used to highlight the alterations in the revised version.

Reviewer #3:

The authors have adequately addressed my previous comments. I only suggest the following:

In response to several of my pervious comments regarding

- 1) type (i.e. serologic, PCR or point-of-care / self-administered) and timing of previous tests,
- 2) the study population (i.e. how many children were seen in total in the 14 hospitals during the serosurvey period? What was the response rate? Were there any differences in characteristics between the people who were included and who were not included in the serosurvey?), and
- 3) source population / catchment area

The authors state that this information is / was not available to them. I understand the challenges faced by the authors to collect this information, but would recommend to include and discuss the lack of information in these respects as further potential limitations of the study.

Reply: We understand your recommendation and have added the following passages to the section on Strengths and Limitations:

Regarding 1):

'As the questionnaire did not specify the timing or applied method of the preceding test (i.e. serologic, PCR or point-of-care / self-administered) and also not the timing of previous tests, the absolute estimate of the underdetection ratios might be biased. With regard to change over time, however, bias will only occur if the number of PCR confirmations per antigen tests changed.'

Regarding 2):

'Unfortunately, we could not validate external validity on the level of the recruiting study sites because we do not have information on the number of admitted children and comparative data regarding recruited and non-recruited patients.'

Regarding 3):

'It is hard to define the catchment areas of the individual study sites. Therefore, we could not account for other determinants such as average socioeconomic status in the catchment area of each hospital in the analysis.'

Thank you very much for your support!

We hope that the manuscript, in its present form, is acceptable for publication in nature communications as a research article.

Respectfully,

Rüdiger von Kries,

for all authors